# Experiment—Simulation Comparison in Liquid Filling Process Driven by Capillarity

**DOI:** 10.3390/mi13071098

**Published:** 2022-07-12

**Authors:** Wei Hua, Wei Wang, Weidong Zhou, Ruige Wu, Zhenfeng Wang

**Affiliations:** Singapore Institute of Manufacturing Technology, Agency for Science, Technology and Research (A*STAR), Singapore 138669, Singapore; wwang@simtech.a-star.edu.sg (W.W.); zhou_weidong@simtech.a-star.edu.sg (W.Z.); rgwu@simtech.a-star.edu.sg (R.W.); zfwang@simtech.a-star.edu.sg (Z.W.)

**Keywords:** capillarity, filling process, Bosanquet equation, model modifications

## Abstract

This paper studies modifications made to the Bosanquet equation in order to fit the experimental observations of the liquid filling process in circular tubes that occurs by capillary force. It is reported that there is a significant difference between experimental observations and the results predicted by the Bosanquet equation; hence, it is reasonable to investigate these differences intensively. Here, we modified the Bosanquet equation such that it could consider more factors that contribute to the filling process. First, we introduced the air flowing out of the tube as the liquid inflow. Next, we considered the increase in hydraulic resistance due to the surface roughness of the inner tube. Finally, we further considered the advancing contact angle, which varies during the filling process. When these three factors were included, the modified Bosanquet equation was well correlated with the experimental results, and the R square—which indicates the fitting quality between the simulation and the experiment—significantly increased to above 0.99.

## 1. Introduction

In microfluidics, one of the most interesting phenomena is the capillarity. Due to the small sizes of components, generally in the sub millimeter or even smaller scale, the capillarity is very strong and is widely employed in various applications, such as capillary pumps [1,2,3] and valves [4,5,6,7], gas debubbling [8], and other applications [9]. Differently to external operations based on instruments, the capillarity is an inherent phenomenon of liquids and can be employed independently. Moreover, it can cooperate with external operations to fulfill various tasks if its features are fully utilized. Due to its importance and contributions, it must be well studied and evaluated in microfluidic flow manipulations, controls, and chip designs.

One famous capillary phenomenon is the self-filling process of liquids in tubes. If the tube is vertically positioned, the liquid will climb up through the tube to a certain height [10] and then stop, and the height can be used to calculate the static contact angle of the liquid. For a horizontally positioned tube, the liquid will fill continually up to the end of the tube. Equations have been developed to describe such filling processes. The Lucas–Washburn equation expresses the filling process without consideration of the liquid density, and it has the disadvantage that the initial speed of the liquid flow is infinitely large. The Bosanquet equation includes the liquid density, and its initial speed decreases to a finite value [11], but it is still restricted in the physical sense that the initial acceleration of the liquid flow is infinitely large. Moreover, it has been found that there are significant differences between experimental observations and the Bosanquet equation [11]. Some modifications of the Boltzmann equation are necessary for better correlations with experimental data.

This paper addresses the modification of the Bosanquet equation so that it can fit experimental data significantly better. First, the gas in the tube is considered; when the liquid is filling the tube, the gas inside is pushed out, which will cause some hydraulic resistance to the liquid. Next, the surface roughness inside the tube will produce extra hydraulic resistance [12], which will make the filling speed slower. Finally, since the advancing contact angle changes during the filling process [13], a variable advancing contact angle is introduced. Differential equations considering these factors are developed, and the fourth order of the Runge–Kutta method is employed to solve the equations numerically. The contributions of the factors in the liquid filling process are discussed in detail.

## 2. Materials and Methods

The Bosanquet equation comes from the following differential equation that governs the liquid filling process in a circular channel by capillary force [11]:(1)$$\rho \frac{d\left(lu\right)}{dt}+RAl\frac{dl}{dt}=\frac{\gamma pcos\theta }{A}$$
where ρ, γ, and θ are the liquid density, the surface tension, and the advancing contact angle, respectively, and p and A are the channel’s cross-sectional perimeter and area, respectively. The advancing contact angle is used because the liquid is moving, and the advancing contact angle is usually larger than the static contact angle in the hydrophilic channel, and l and u are the liquid filling distance and speed, respectively. Here, R is defined as the normalized hydraulic resistance ($R=128\mu /\left(\pi d^{4}\right)$ for a circular tube), and the hydraulic resistance of the channel is Rl. Table 1 lists the terms and definitions for convenience. By solving the differential equation with the boundary conditions:(2)$$\underset{t\to 0}{\lim}l=\underset{t\to 0}{\lim}lu=0$$

The Bosanquet equation is obtained as:(3)$$l=\frac{1}{2}\sqrt{\frac{d\sigma cos\theta }{\mu }}\sqrt{t-\frac{\rho d^{2}}{32\mu }\left(1-e^{-\frac{32\mu }{\rho d^{2}}t}\right)}$$

It was interesting to find that the initial speed $\underset{t\to 0}{\lim}u=\sqrt{p\gamma cos\theta /\left(A\rho \right)}$. This meant that the liquid’s acceleration was infinitely large at t = 0, which conflicted with the common-sense principles of physics. To solve this singularity problem, we introduced the gas flow in the tube. As the liquid fills the tube, the gas inside the tube will flow out, and we suppose that the speed of the liquid filling is the same as the gas flowing out, as illustrated in Figure 1. Moreover, the surface roughness of the inner tube is expected to generate a larger pressure drop than the case of a smooth tube [12]; hence, the surface roughness effect was considered here. Furthermore, the advancing contact angle varies with the filling speed, and this variation should be considered. As a result, Equation (1) could be modified as:(4)$$\rho \frac{d\left(lu\right)}{dt}+\rho_{g}\frac{d\left(\left(L-l\right)u\right)}{dt}+\alpha RAl\frac{dl}{dt}+R_{g}A\left(L-l\right)\frac{dl}{dt}=\frac{\gamma pcos\theta }{A}$$

Here, the parameter α is related to the surface roughness without units, and α ≥ 1. When α = 1, this means that the surface roughness is not considered. The subscript g refers to the gas. The term $R_{g}A\left(L-l\right)dl/dt$ is for the hydraulic resistance of gas, and the term $\rho_{g}d\left(\left(L-l\right)u\right)/dt$ considers the gas density effect in the gas flow. At first, we took the advancing contact angle as a fixed value. By integrating Equation (4), we obtained:(5)$$\left(\rho -\rho_{g}\right)lu+\frac{\alpha R-R_{g}}{2}Al^{2}+\rho_{g}Lu+R_{g}ALl=\frac{\gamma pcos\theta }{A}t$$

Determining the limit by Equation (5):(6)$$\underset{t\to 0}{\lim}\left[\left(\rho -\rho_{g}\right)lu+\frac{\alpha R-R_{g}}{2}Al^{2}+\rho_{g}Lu+R_{g}ALl\right]=\underset{t\to 0}{\lim}\frac{\gamma pcos\theta }{A}t=0$$

We know that $\underset{t\to 0}{\lim l}=0$ and $\rho_{g}L>0$; so, $\underset{t\to 0}{\lim u}=0$. Therefore, the singularity problem is solved by the introduction of gas flowing out of the tube.

Since there is no analytical solution of Equation (5) in which u=dl/dt, a numerical method based on the fourth order of the Runge–Kutta method was used. We also employed the same dynamic contact angle values used in [11], which were fixed at 85° for the case of a 0.5 mm diameter and 83° for the case of a 1.15 mm diameter in that paper. The simulation results and their comparisons with the experimental results are presented in the next section.

Next, we considered the variation in the advancing contact angle and solved Equation (4) directly. It is reported that the advancing contact angle can be expressed as [13]:(7)$$cos\theta =cos\theta_{0}-2\left(1+cos\theta_{0}\right)\sqrt{C_{a}}$$
where $\theta_{0}$ is the static contact angle and $C_{a}=\mu u/\gamma$ is the capillary number. The static contact angle can be obtained by measuring the height rise in the vertical tube, $\theta_{0}=cos^{-1}\left(dh\rho g/\left(4\gamma \right)\right)$. The height rises were 9.3 mm and 4.9 mm in the tubes with 0.5 and 1.15 mm diameters, respectively; hence, their related static contact angles were approximately 81° and 79°, respectively, using the parameters in Table 2—which were obtained from websites using the experimental temperature of 23 °C. Submitting Equation (7) into Equation (4), it gave:(8)$$\left(\rho -\rho_{g}\right)\frac{d\left(lu\right)}{dt}+\left(\alpha R-R_{g}\right)Al\frac{dl}{dt}+\rho_{g}L\frac{du}{dt}+R_{g}AL\frac{dl}{dt}=\frac{\gamma p}{A}\left(cos\theta_{0}-2\left(1+cos\theta_{0}\right)\sqrt{\frac{\mu }{\gamma }}\sqrt{u}\right)$$

From Equation (7), it can be observed that the advancing contact angle is proportional to the liquid flowing speed. At the beginning stage of filling, the liquid speed is higher; hence, the advancing contact angle is larger. As time goes on, the filling speed will be slower and slower, which indicates that the advancing contact angle will reduce accordingly. By solving Equation (8) directly using the fourth order of the Runge–Kutta method, the filling characteristics can be obtained and are presented in the next section.

## 3. Experiment and Simulation Comparisons

It has been reported that there are significant differences between experimental data and the Bosanquet equation [11]—as illustrated in Figure 2, which also includes the latest simulation results. In the figure, Figure 2a is for a tube 0.5 mm in diameter and 90 mm in length, while Figure 2b is for a tube 1.15 mm in diameter and 75 mm in length. The tubes were made of glass, and they were marked at fixed intervals of 5 mm. When the dyed water passed through each mark, the time was recorded by a handphone video. The time interval between two continual pictures was 1/30 s. From the pictures, the times taken to reach all the marks were obtained accordingly. As the shot rate was not high, experimental data uncertainties existed—especially for the tube 1.15 mm in diameter, in which the filling speed was significantly faster. Model 1 represents the case in which only the gas in tube was considered; Model 2 indicates the case in which both the gas in the tube and the surface roughness of the inner tube were considered, while Model 3 considered all the gas in the tube, the surface roughness of the inner tube, and the variable advancing contact angle. The R square, which indicates the curve fitting performance, was also given. Since the experiment was conducted at a lab temperature of 23 °C with water, the property parameters of water and air at this temperature are listed in Table 2.

In Figure 2a, it can be observed that the Bosanquet equation was significantly higher than the experimental data, and that the R square was only 0.8889. By introducing the gas in the tube, the differences were smaller, and the R square increased to 0.9316. When the surface roughness of the tube was considered, the differences were significantly reduced, with the R square increasing up to 0.9882. Here, the value of α was determined by maximizing the R square. Figure 3 illustrates the relationship between the R square and α for Model 2. When α equaled 1.25, the R square obtained its maximum value of 0.9882; this means that if the hydraulic resistance increases by 25% due to the surface roughness of the inner tube, Model 2 can fit the experimental data best. Similarly, for the inclusion of the variable of the advancing contact angle, as illustrated by Model 3, the maximum R square was obtained at 0.9929. It is shown that by modification of the Bosanquet equation, the differences between the experimental data and the simulation could be significantly reduced. Figure 2b shows the results from a tube with a larger diameter and shorter length. It can be observed that the filling speed was much faster than that in Figure 2a, and this was taken account as the reduction in hydraulic resistance—which is inversely proportional to the fourth order of the tube diameter—was more significant than the reduction in the capillary pressure, which was inversely proportional to the tube diameter only. It was found that all the models in Figure 2b, including the Bosanquet equation, fit the experimental data better than those in Figure 2a. One explanation is that when the tube size increases, the contribution of the surface roughness is less [12]; hence, the results are closer to the experimental data. When α is 1.12, Model 2 will get its maximum R square, which is significantly less than the 1.25 shown in Figure 3 for Model 2 in Figure 2a. It is interesting to note that the R square in Model 2 was even larger than that in Model 3. Since the experimental data were obtained from the handheld cellular phone video (30 shots per second)—and due to the filling speed in Figure 2b being significantly faster than that in Figure 2a—the experimental data may not be accurate enough, and it may not be possible to determine which model—Model 2 or 3—is more suitable for Figure 2b. Generally, both Models 2 and 3 had high R squares and fit the experimental data quite well.

At the rear region of the tube in Figure 2, where the filling process ends, it can be noticed that the filling speed in Model 3 was faster than that in Model 2. As the filling speed was significantly reduced in the rear region, and the advancing contact angle was reduced accordingly—as shown by Equation (7)—this made the driving pressure increase, and hence the reduction in the filling speed was less. As a result, the curve of Model 3 surpassed the curve of Model 2 in terms of the filling distance, as illustrated in both Figure 2a,b.

From the Bosanquet equation, the initial filling speed of a circular tube is $2\sqrt{\gamma cos\theta_{0}/\left(d\rho \right)}$. By submitting the parameters in Table 2 into the equation, we obtained initial filling speeds of 225 and 175 mm/s for tubes of 0.5 and 1.15 mm in diameter, respectively. The initial speeds were also the maximum speeds in the filling processes. Here, we are more interested in the speeds of the three simulation models, and Figure 4 illustrates their results. It is observed that their initial speeds—both in Figure 4a,b—were 0, which is reasonable from the viewpoint of physics.

In the beginning stage of Figure 4a, the filling speeds of all the models increased very rapidly. There was little difference between Models 1 and 2, because at the beginning stage the filling distance was very small; thus, the contribution of the hydraulic resistance was small too, and the roughness effect on the hydraulic resistance was even smaller. Model 1 obtained its maximum speed of 136.9 mm/s at 1.55 ms, while for Model 2, it was 136 mm/s at 1.45 ms. The small difference was caused by the hydraulic resistance. For Model 3, the maximum speed was 114.5 mm/s at 1.05 ms. Since the speed was very high at the beginning stage, the advancing contact angle was also large, which made the driving pressure small, and hence the maximum speed was lower and arrived earlier. When the maximum speeds of the three models are compared with the initial speed of the Bosanquet equation, they are significantly smaller. At the rear stage, the speed of Model 1 was the largest, followed by that of Models 3 and 2. Model 1 was the fastest because it did not have roughness-caused hydraulic resistance. Due to the reduction in speed, the advancing contact angle also decreased, which made the capillary pressure increase, and thus caused the speed of Model 3 to be larger than that of Model 2.

For the tube 1.15 mm in diameter, the filling speed is illustrated in Figure 4b. At the beginning stage, the maximum speeds of Models 1, 2, and 3 were 155.3 mm/s at 3.13 ms, 155 mm/s at 3.12 ms, and 134 mm/s at 2.33 ms, respectively. These values are larger than those shown in Figure 4a, but the times are delayed. This was mainly caused by the tube diameter, which made a very large contribution to the hydraulic resistance. For the Bosanquet equation, the smaller the diameter, the larger the initial speed—which is opposite to the present simulation results. At the rear stage, it is interesting to find that the speed of Model 3 was even faster than that of Model 1, which is different from the results shown in Figure 4a. This can be explained as the surface roughness contribution on hydraulic resistance being lower due to a larger tube diameter [12]; thus, the increase in capillary pressure due to the speed reduction could lead to an increase in hydraulic resistance caused by surface roughness. The contribution of the surface roughness to the hydraulic resistance can be evaluated by the difference between Model 1 and Model 2, in which the speed was decreased by approximately 1 mm/s at the rear stage.

Figure 5 illustrates the relationship between the advancing contact angle and time, which is obtained directly from Equation (7). Due to the rapid increase in the filling speed at the beginning stage, as illustrated in Figure 4, the advancing contact angle also increased rapidly. The maximum angles for Figure 5a,b were 86.06° at 1.05 ms and 84.66° at 2.33 ms, respectively. Thereafter, the angles decreased quickly and then slowly reduced—just like the filling speeds in Figure 4. It can be noticed that in Figure 2a and Figure 4a, the advancing contact angle used in Models 1 and 2 was fixed as 85°, but for Model 3, the advancing contact angle is as illustrated by Figure 5a. Since at the rear stage the angle used in Model 3 was smaller than that used in Model 2, it is reasonable that the speed in Model 3 was faster than that in Model 2 at the rear stage—as illustrated by Figure 4a. For the cases of Figure 2b and Figure 4b, the advancing contact angle was fixed as 83° for Models 1 and 2, while Model 3 employed the angle shown in Figure 5b, in which at the rear stage the angle was also smaller than 83°. This can be used to explain the speed of Model 3 being faster than that of Model 2 at the rear stage, as illustrated in Figure 4b.

## 4. Conclusions

The capillary filling process of liquids was investigated in detail in the present paper. A previous publication [11] showed that the filling distance predicted by the Bosanquet equation is significantly larger than that in experimental observations. By modification of the Bosanquet equation, including considerations of the gas flowing in the tube, hydraulic resistance increases due to the surface roughness of the inner tube, and the variable advancing contact angle, the simulation and the experiment were well correlated, with a R square above 0.99.

The Bosanquet equation has a non-zero initial speed, which is in conflict with common-sense physics principles, since the initial acceleration of a liquid cannot be infinitely large. By introducing the concept of the gas flowing inside the tube, such a singularity was removed, the initial speed became zero, and the filling distance was slightly reduced. With the inclusion of the surface roughness, the hydraulic resistance increased and the filling distance was significantly reduced. When the variable advancing contact angle was considered, the filling speed at the rear stage was larger, due to the advancing contact angle being less; hence, the capillary pressure as well as the filling speed was larger at that stage. 

Compared with the Bosanquet equation, the modified model is more consistent with the actual situation of the filling process. Therefore, it is reasonable that the simulation results are closer to the experimental data.

## Figures and Tables

**Figure 1 micromachines-13-01098-f001:**
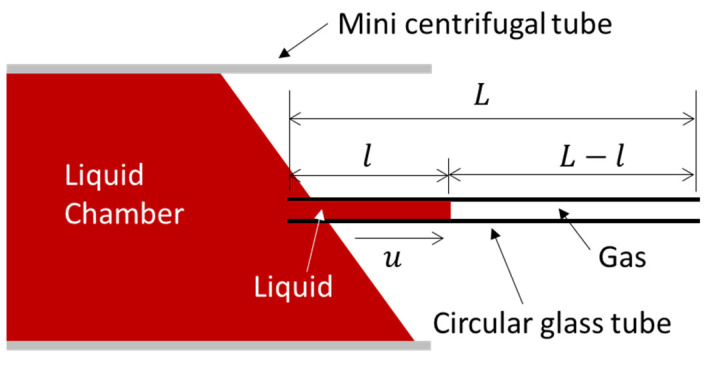
Liquid filling process in a circular channel. When a liquid enters the channel, the gas inside is pushed out.

**Figure 2 micromachines-13-01098-f002:**
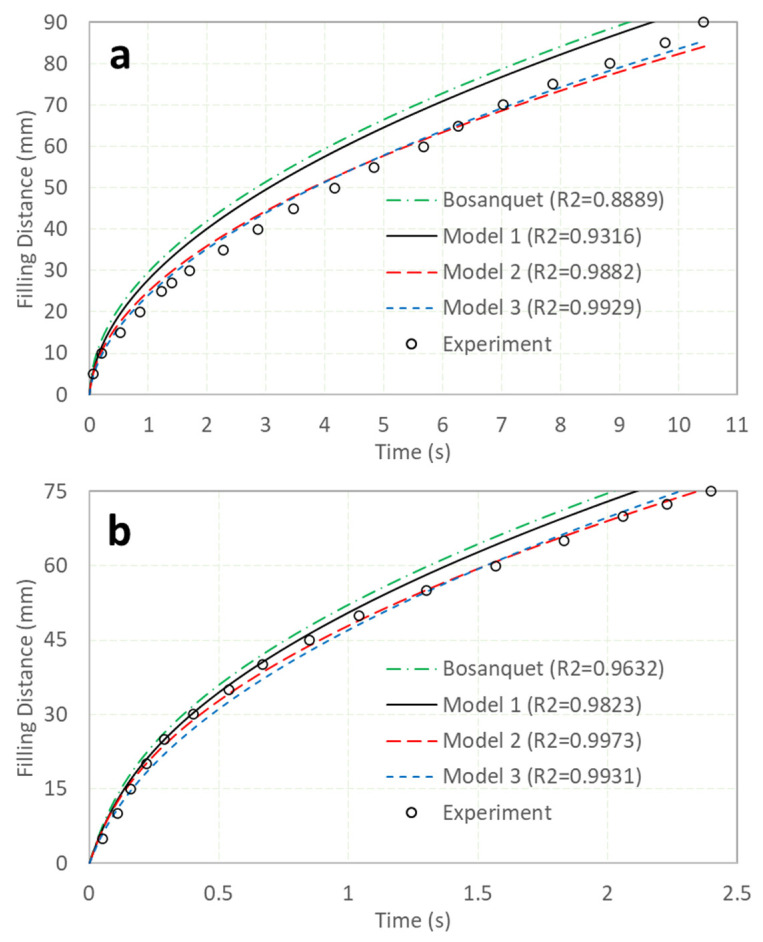
Experiment and simulation comparisons. The data from the experiment and the Bosanquet equation are from [11] directly; Model 1 considers gas in the tube only, Model 2 considers both the gas in the tube and the inner tube surface roughness, and Model 3 considers all the gas in the tube, the inner tube surface roughness, and the variable advancing contact angle. Here, (**a**) and (**b**) indicate tubes 0.5 and 1.15 mm in diameter, respectively.

**Figure 3 micromachines-13-01098-f003:**
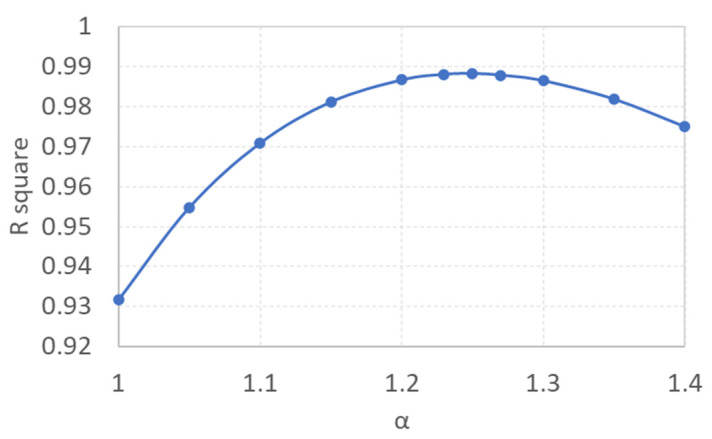
R square versus *α* for Model 2 in Figure 2a. When *α* = 1, there is no surface roughness considered, and it is equivalent to Model 1. When *α* = 1.25, R square gets its maximum value of 0.9882.

**Figure 4 micromachines-13-01098-f004:**
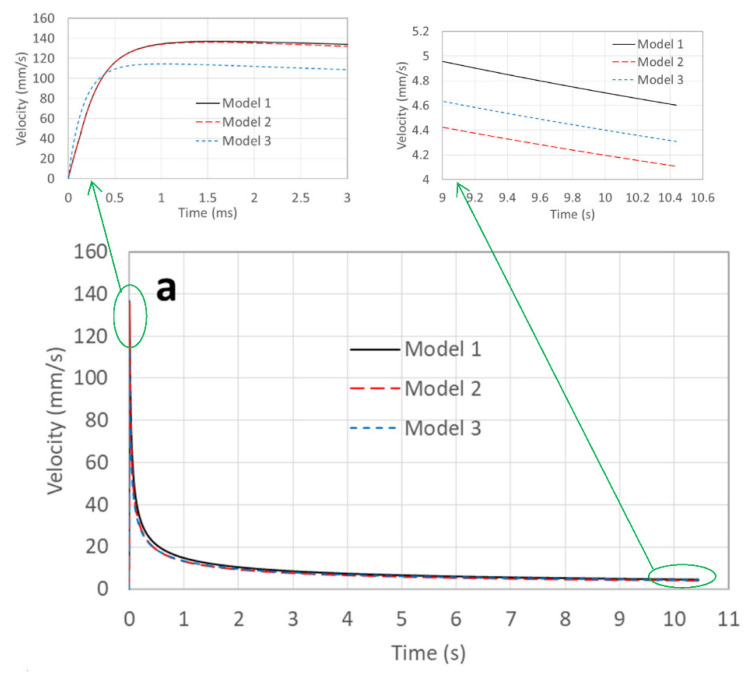
Filling speed versus time in the tube filling process. Here, (**a**) and (**b**) indicate tubes 0.5 and 1.15 mm in diameter, respectively.

**Figure 5 micromachines-13-01098-f005:**
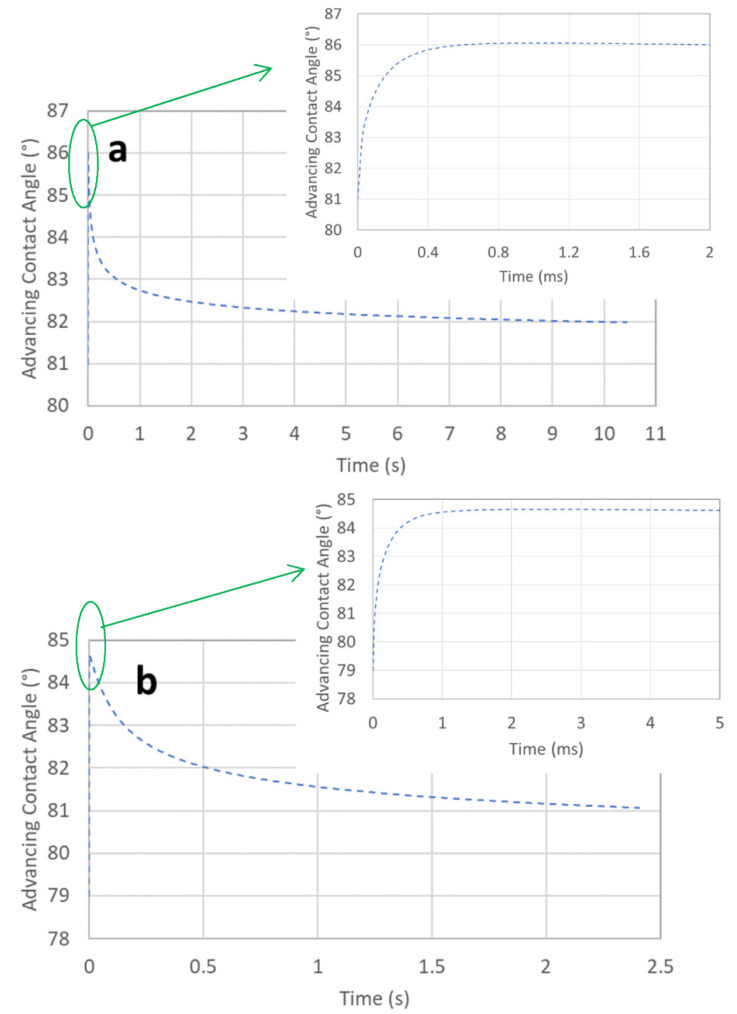
Advancing contact angle versus time in Model 3. Here, (**a**) and (**b**) indicate tubes of 0.5 and 1.15 mm in diameter, respectively.

**Table 1 micromachines-13-01098-t001:** Terms and definitions.

A	Cross-sectional area of channel (m^2^)
$C_{a}$	Capillary number μu/γ (no unit)
d	Diameter of circular channel (m)
l	Liquid filling distance (m)
L	Channel length (m)
p	Cross-sectional perimeter of channel (m)
R	Normalized hydraulic resistance of liquid (Pa.s/m^4^)
$R_{g}$	Normalized hydraulic resistance of gas (Pa.s/m^4^)
t	Filling time (s)
u	Filling velocity (m/s)
α	Surface roughness contribution α≥1 (no unit)
ρ	Liquid density (kg/m^3^)
$\rho_{g}$	Gas density (kg/m^3^)
θ	Advancing contact angle (rad)
$\theta_{0}$	Static contact angle (rad)
μ	Viscosity coefficient of liquid (Pa.s)
$\mu_{g}$	Viscosity coefficient of gas (Pa.s)
γ	Surface tension of liquid (N/m)

**Table 2 micromachines-13-01098-t002:** Water and air properties at a lab temperature of 23 °C.

	Liquid (Water)	Gas (Air)
Density (kg/m^3^)	997.1	1.192
Viscosity (Pa.s)	0.9096 × 10^−3^	1.835 × 10^−5^
Surface tension (N/m)	0.0723	NA
Static contact angle (°)	81 (Figures 2a, 4a and 5a)	NA
	79 (Figures 2b, 4b and 5b)	

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
