# Peer review of "Experiment—Simulation Comparison in Liquid Filling Process Driven by Capillarity"

_micromachines, 2022, doi:10.3390/mi13071098_

Round 1
Reviewer 1 Report
This paper is about the study of liquid filling inside a tube by using capillarity. Authors compare the experimental results with the derived data from Bosanquet equation. researchers modified the Bosanquet equation to be more suitable based on their previous work.
The manuscript is well written but needs to be reorganized. The applicated methodology is good. Furthermore, the paper should have full consideration to be published in micromachines journal after making some necessary modifications.
1 1) Lines 47-50: this paragraph seems to be good in the conclusion rather than in the introduction.
2 2) Line 60: Bad notation R instead Rl.
3 3) Line 61: The boundary conditions should be listed as equations.
4 4) Table 1: Used units for each parameter should be added.
5 5) Line 67: Why the acceleration has an infinite value at t=0?
6 6) Lines 79-81: the expressions should be listed as equations.
7 7) Which equation is solved numerical, eq 3 or 4? Equation 4 isn’t a differential equation.
8 8) Authors didn’t explain how the experimental part is done.
9 9) All figures: the caption isn’t appropriate and too long. Please eliminate the unnecessary information.
1 10) Authors need to refer figure 1 in the text.
1 11) The results of velocity in the discussion section should be moved to precedent section. Results and discussion sections is better to be mixed.
1 12) How we can choose between Model 2 or 3? It seems that the diameter of the tube influence the model performance, at which diameter we can applicate Model 2 or 3?
Author Response
We have replied to Reviewer 1 in a Word file: Answer to Reviewer 1.docx. Please refer to the attached file. (Since some equations exist in the answers)

Reviewer 2 Report
Major Comments:
1. The formatting of descriptions for the equations, corresponding boundary conditions and the corresponding subscripts should be fixed. (i.e., Page 2 Line 61; Page 3 Line 66, 79, and 90 etc.)
2. Please proofread. I have noticed numerous incorrect sentences.
3. All three models are being compared with experimental data for validation. The experimental data are from previous literature published by the authors of this manuscript (Reference 11). However, the experimental uncertainty, standard deviation or errors have not been specified for the experimental data. As the differences between the predictions of these models are only modest, there is high probability these predictions would have been within the uncertainty margin of the experimental data. The authors should address this concern as the cited literature (Reference 11) for the experimental data are also from their group.
Author Response
Question 1: The formatting of descriptions for the equations, corresponding boundary conditions and the corresponding subscripts should be fixed. (i.e., Page 2 Line 61; Page 3 Line 66, 79, and 90 etc.)
Answer 1: The formatting problems are corrected thoroughly (caused by auto transfer from original paper to current format). We are sorry for the troubles caused.
Question 2: Please proofread. I have noticed numerous incorrect sentences.
Answer 2: The whole paper has been checked and the incorrect sentences are rewritten.
Question 3: All three models are being compared with experimental data for validation. The experimental data are from previous literature published by the authors of this manuscript (Reference 11). However, the experimental uncertainty, standard deviation or errors have not been specified for the experimental data. As the differences between the predictions of these models are only modest, there is high probability these predictions would have been within the uncertainty margin of the experimental data. The authors should address this concern as the cited literature (Reference 11) for the experimental data are also from their group.
Answer 3: It is a good question. We agree with the reviewer on the experimental uncertainty, especially for Figure b (diameter of 1.15 mm), because in that tube the filling speed is much faster than that of Figure a (diameter of 0.5 mm). In Reference 11, we used HP to take videos, which has shot rate of 30 shots per second. For more accurate experimental data, high speed cameras should be employed, which is our next plan. For clearance, some experimental introductions are added In lines 113-118.
“The tubes are made of glass, and they are marked with a fixed interval of 5 mm. When the dyed water passes through each mark, the time is recorded by handphone video. The time interval between two continual pictures is 1/30 s. From the pictures the times to reach all the marks are obtained accordingly. Due to the shot rate is not high, experimental data uncertainties exist, especially for the tube of 1.15 mm in diameter in which the filling speed is faster significantly.”

Round 2
Reviewer 2 Report
The authors have addressed all of my comments.